# Adaptive Importance Sampling for Finite-Sum Optimization and Sampling with Decreasing Step-Sizes

**Ayoub El Hanchi**
McGill University
ayoub.elhanchi@mail.mcgill.ca

**David A. Stephens**
McGill University
david.stephens@mcgill.ca

## Abstract

Reducing the variance of the gradient estimator is known to improve the convergence rate of stochastic gradient-based optimization and sampling algorithms. One way of achieving variance reduction is to design importance sampling strategies. Recently, the problem of designing such schemes was formulated as an online learning problem with bandit feedback, and algorithms with sub-linear *static* regret were designed. In this work, we build on this framework and propose *Avare*, a simple and efficient algorithm for adaptive importance sampling for finite-sum optimization and sampling with decreasing step-sizes. Under standard technical conditions, we show that *Avare* achieves $\mathcal{O}(T^{2/3})$ and $\mathcal{O}(T^{5/6})$ *dynamic* regret for SGD and SGLD respectively when run with $\mathcal{O}(1/t)$ step sizes. We achieve this dynamic regret bound by leveraging our knowledge of the dynamics defined by the algorithm, and combining ideas from online learning and variance-reduced stochastic optimization. We validate empirically the performance of our algorithm and identify settings in which it leads to significant improvements.

## 1   Introduction

Functions $f : \mathbb{R}^d \to \mathbb{R}$ of the form:

$$f(x) = \sum_{i=1}^{N} f_i(x) \tag{1}$$

are prevalent in modern machine learning and statistics. Important examples include the empirical risk in the empirical risk minimization framework, [1] or the log-posterior of an exchangeable Bayesian model. When $N$ is large, the preferred methods for solving the resulting optimization or sampling problem usually rely on stochastic estimates of the gradient of $f$, using variants of stochastic gradient descent (SGD) [1] or stochastic gradient Langevin dynamics (SGLD) [2]:

$$x_{t+1}^{SGD} = x_t^{SGD} - \alpha_t N \nabla f_{I_t}(x_t^{SGD}) \tag{2}$$

$$x_{t+1}^{SGLD} = x_t^{SGLD} - \alpha_t N \nabla f_{I_t}(x_t^{SGLD}) + \xi_t \qquad \xi_t \sim \mathcal{N}(0, 2\alpha_t) \tag{3}$$

where $\{\alpha_t\}_{t=1}^{T}$ is a sequence of step-sizes, and the index $I_t$ is sampled uniformly from $[N]$, making $N \nabla f_{I_t}(x)$ an unbiased estimator of the gradient of $f$. We use $\{x_t\}_{t=1}^{T}$ to refer to either sequence when we do not wish to distinguish between them. It is well known that the quality of the answers given by these algorithms depends on the (trace of the) variance of the gradient estimator, and considerable efforts have been made to design methods that reduce this variance.

In this paper, we focus on the importance sampling approach to achieving variance reduction. At each iteration, the algorithm samples $I_t$ according to a specified distribution $p^t$, and estimates the gradient using:

$$\hat{g}^t := \frac{1}{p_{I_t}^t} \nabla f_{I_t}(x_t) \tag{4}$$

It is immediate to verify that $\hat{g}^t$ is an unbiased estimator of $g^t := \nabla f(x_t)$. By cleverly choosing the distributions $p^t$, one can achieve significant variance reduction (up to a factor of N) compared to the estimator based on uniform sampling. Unfortunately, computing the variance-minimizing distributions at each iteration requires the knowledge of the Euclidean norm of all the individual gradients $g_i^t := \nabla f_i(x_t)$ at each iteration, making it unpractical [3, 4].

Many methods have been proposed that attempt to construct sequences of distributions $\{p^t\}_{t=1}^T$ that result in efficient estimators [3, 4, 5, 6, 7, 8, 9]. Of particular interest to us, the task of designing such sequences was recently cast as an online learning problem with bandit feedback [10, 11, 12, 13]. In this formulation, one attempts to design algorithms with sub-linear *expected* static regret, which is defined as:

$$\text{Regret}_S(T) := \sum_{t=1}^T c_t(p^t) - \min_{p \in \Delta} \sum_{t=1}^T c_t(p)$$

where $\Delta$ denotes the probability simplex in $\mathbb{R}^N$, and $c_t(p)$ is the trace of the covariance matrix of the gradient estimator (4), which is easily shown to be:

$$c_t(p) := \sum_{i=1}^N \frac{1}{p_i} \left\| g_i^t \right\|_2^2 - \left\| g^t \right\|_2^2 \tag{5}$$

Note that the second term cancels in the definition of regret, and we omit it in the rest of our discussion. In this formulation, and to keep the computational load manageable, one has only access to partial feedback in the form of the norm of the $I_t^{th}$ gradient, and not to the complete cost function (5). Under the assumption of uniformly bounded gradients, the best result in this category can be found in [12] where an algorithm with $\tilde{O}(T^{2/3})$ static regret is proposed. A more difficult but more natural performance measure that makes the attempt to approximate the optimal distributions explicit is the *dynamic* regret, defined by:

$$\text{Regret}_D(T) := \sum_{t=1}^T c_t(p^t) - \sum_{t=1}^T \min_{p \in \Delta} c_t(p) \tag{6}$$

Guarantees with respect to the expected dynamic regret are more difficult to obtain, and require that the cost functions $c_t(p)$ do not change too rapidly with respect to some reasonable measure of variation. See [14, 15, 16, 17, 18] for examples of such measures and the corresponding regret bounds for general convex cost functions.

In this work, we propose *Avare*, an algorithm that achieves sub-linear dynamic regret for both SGD and SGLD when the sequence of step-sizes $\{\alpha_t\}_{t=1}^T$ is decreasing. The name *Avare* is derived from adaptive variance minimization. Specifically, our contributions are as follows:

- We show that *Avare* achieves $\mathcal{O}(T^{2/3})$ and $\mathcal{O}(T^{5/6})$ dynamic regret for SGD and SGLD respectively when $\alpha_t$ is $\mathcal{O}(1/t)$.

- We propose a new mini-batch estimator that combines the benefits of sampling without replacement and importance sampling while preserving unbiasedness.

- We validate empirically the performance of our algorithm and identify settings in which it leads to significant improvements.

We would like to point out that while the decreasing step size requirement might seem restrictive, we argue that for SGD and SGLD, it is the right setting to consider for variance reduction. Indeed, it is well known that under suitable technical conditions, both algorithms converge to their respective solutions exponentially fast in the early stages. Variance reduction is primarily useful at later stages when the noise from the stochastic gradient dominates. In the absence of control variates, one is forced to use decreasing step-sizes to achieve convergence. This is precisely the regime we consider.

## 2 Related work

It is easy to see that the cost functions (5) are convex over the probability simplex. A celebrated algorithm for convex optimization over the simplex is entropic descent [19], an instance of mirror descent [20] where the Bregman divergence is taken to be the relative entropy. A slight modification of this algorithm for online learning is the EXP3 algorithm [21] which mixes the iterates of entropic descent with a uniform distribution to avoid the assignment of null probabilities. See [22, 23, 24] for a more thorough discussion of online learning (also called online convex optimization) in general and variants of this algorithm in particular.

Since we are working over the simplex, the EXP3 algorithm is a natural choice. This is the approach taken in [10] and [11], although strictly speaking neither is able to prove sub-linear static regret bounds. The difficulty comes from the fact that the norm of the gradients of the cost functions (5) explode to infinity on the boundary of the simplex. This is amplified by the use of stochastic gradients which grow as $1/p_{min}^5$ in expectation, making it very difficult to reach regions near the boundary. Algorithms based on entropic descent for dynamic regret problems also exist, including the fixed share algorithm and projected mirror descent [25, 26, 27, 28]. Building on these algorithms, and by artificially making the cost functions strongly-convex to allow the use of decreasing step-sizes, we were only able to show $\tilde{O}(T^{7/8})$ dynamic regret using projected mirror descent for SGD with $O(1/t)$ decreasing step sizes and uniformly bounded gradients.

The approach taken in [12] is qualitatively different and is based on the follow-the-regularized-leader (FTRL) scheme. By solving the successive minimization problems stemming from the FTRL scheme analytically, the authors avoid the above-mentioned issue of exploding gradients. The rest of their analysis relies on constructing an unbiased estimate of the cost functions (5) using only the partial feedback $\left\|g_{I_t}^t\right\|_2^2$, and probabilistically bounding the deviation of the estimated cost functions from the true cost functions using a martingale concentration inequality. The final result is an algorithm that enjoys an $\tilde{O}(T^{2/3})$ static regret bound.

Our approach is similar to the one taken in [12] in that we work directly with the cost functions and not their gradients to avoid the exploding gradient issue. Beyond this point however, our analysis is substantially different. While we are still working within the online learning framework, our analysis is more closely related to the analysis of variance-reduced stochastic optimization algorithms that are based on control variates. In particular, we study a Lyapunov-like functional and show that it decreases along the trajectory of the dynamics defined by the algorithms. This yields simpler and more concise proofs, and opens the door for a unified analysis.

## 3 Algorithm

Most of the literature on online convex optimization with dynamic regret relies on the use of the gradients of the cost functions [14, 15, 16, 17, 18]. However, as explained in the previous section, such approaches are not viable for our problem. Unfortunately, the regret guarantees obtained from the FTRL scheme for static regret used in [12] do not directly translate into guarantees for dynamic regret.

We start by presenting the high level ideas that go into the construction of our algorithm. We then present our algorithm in explicit form, and discuss its implementation and computational complexity.

### 3.1 High level ideas

The simplest update rule one might consider when working with dynamic regret is a natural extension of the follow-the-leader approach:

$$p^{t+1} := \underset{p \in \Delta}{\operatorname{argmin}} \left\{ c_t(p) \right\} \tag{7}$$

Intuitively, if the cost functions do not change too quickly, then it is reasonable to expect good performance from this algorithm. In our case however, we do not have access to the full cost function. The traditional way of dealing with this problem is to build unbiased estimates of the cost functions using the bandit feedback, and then bounding the deviation of these unbiased estimates from the true cost functions. While this might work, we consider here a different approach based on constructing surrogate cost functions that are not necessarily unbiased estimates of the true costs.

For each $i \in [N]$, denote by $h_i^t$ the last observed gradient of $f_i$ at time $t$, with $h_i^1$ initialized arbitrarily (to 0 for example). We consider the following surrogate cost for all $t \in [T]$:

$$\widetilde{c}_t(p) := \sum_{i=1}^{N} \frac{1}{p_i} \left\| h_i^t \right\|_2^2 \tag{8}$$

As successive iterates of the algorithm become closer and closer to each other, the squared norm of the $h_i^t$s become better and better approximations to the squared norm of the $g_i^t$s, thereby making the surrogate costs (8) accurate approximations of the true costs (5). The idea of using previously seen gradients to approximate current gradients is inspired by the celebrated variance-reduced stochastic optimization algorithm SAGA [29].

We are now almost ready to formulate our algorithm. If we try to directly minimize the surrogate cost functions over the entire simplex at each iteration as in (7), we might end up assigning null or close to null probabilities to certain indices. Depending on how far we are in the algorithm, this might or might not be a problem. In the initial stages, this is clearly an issue since this might only be an artifact of the initialization (notably when we initialize the $h_i^1$ to 0), or an accurate representation of the current norm of the gradient, but which might not be representative later on as the algorithm progresses. On the other hand, in the later stages of the algorithm, the cost functions are nearly identical, so that an assignment of near zero probabilities is a reflection of the true optimal probabilities, and is not necessarily problematic.

The above discussion suggests the following algorithm. Define the $\varepsilon$-restricted probability simplex to be:

$$\Delta(\varepsilon) := \left\{ p \in \mathbb{R}^N \mid p_i \geq \varepsilon, \ \sum_{i=1}^{N} p_i = 1 \right\} \tag{9}$$

And let $\{\varepsilon_t\}_{t=1}^{T}$ be a decreasing sequence of positive numbers with $\varepsilon_1 \leq \frac{1}{N}$. Then we propose the following algorithm:

$$p^t := \operatorname*{argmin}_{p \in \Delta(\varepsilon_t)} \{\widetilde{c}_t(p)\} \tag{10}$$

Our theoretical analysis in Section 4 suggests a specific decay rate for the sequence $\{\varepsilon_t\}_{t=1}^{T}$ that depends on the sequence of step-sizes $\{\alpha_t\}_{t=1}^{T}$ and whether SGD or SGLD is run.

## 3.2 Explicit form

Equation (10) defines our sequence of distribution $\{p^t\}_{t=1}^{T}$, but the question remains whether we can solve the implied optimization problems efficiently. In this section we answer this question in the affirmative, and provide an explicit algorithm for the computation of the sequence $\{p^t\}_{t=1}^{T}$.

We state our main result of this section in the following lemma. The proof can be found in appendix A.

**Lemma 1.** *Let $\{a_i\}_{i=1}^{N}$ be a non-negative set of numbers where at least one of the $a_i$s is strictly positive, and let $\varepsilon \in [0, 1/N]$. Let $\pi : [N] \to [N]$ be a permutation that orders $\{a_i\}_{i=1}^{N}$ in a decreasing order ($a_{\pi(1)} \geq a_{\pi(2)} \geq \cdots \geq a_{\pi(N)}$). Define:*

$$\rho := \max \left\{ i \in [N] \ \middle| \ a_{\pi(i)} \geq \varepsilon \frac{\sum_{j=1}^{i} a_{\pi(j)}}{1 - (N-i)\varepsilon} \right\} \tag{11}$$

*and:*

$$\lambda := \frac{\sum_{j=1}^{\rho} a_{\pi(j)}}{1 - (N - \rho)\varepsilon} \tag{12}$$

*Then a solution of the optimization problem:*

$$\min_{p \in \Delta(\varepsilon)} \sum_{i=1}^{N} \frac{1}{p_i} a_i^2 \tag{13}$$

*is given by:*

$$p_i^* = \begin{cases} a_i/\lambda & \text{if } i \in \{\pi(1), \dots, \pi(\rho)\} \\ \varepsilon & \text{otherwise} \end{cases} \tag{14}$$

*In the case all $a_i$ are zero, any $p \in \Delta(\varepsilon)$ is a solution.*

In light of Lemma 1, to compute $p^t$ as defined in (10), it is enough to know the value of $\rho_t$ as defined in (11), replacing $a_i$ with $\|h_i^t\|_2$ and $\varepsilon$ with $\varepsilon_t$. Using $\rho_t$ we can then compute $\lambda_t$ using (12), and obtain $p^t$ from (14). It remains to specify an efficient way to perform this computation.

## 3.3 Implementation details

The naive way to perform the above computation is to do the following at each iteration:

- Sort $\{\|h_i^t\|_2\}_{i=1}^N$ in decreasing order.
- Find $\rho_t$ by traversing $(\pi(i))_{i=1}^N$ in increasing order and finding the first $i \in [N]$ for which the inequality in (11) does not hold.
- Explicitly compute the probabilities using (12) and (14).
- Sample from $p^t$ using inverse transform sampling.

This has complexity $O(N \log N)$. In appendix A, we present an algorithm that requires only $O(N)$ vectorized operations, $O(\log^2 N)$ sequential (non-vectorized) operations, and $cN$ memory for small $c$. The algorithm uses three data structures:

- An array storing $\{\|h_i^t\|_2\}_{i=1}^N$ unsorted.
- An order statistic tree storing the pairs $(\|h_i^t\|_2, i)_{i=1}^N$ sorted according to $\|h_i^t\|_2$.
- An array storing $\{\sum_{j=1}^i \|h_{\pi(j)}^t\|_2\}_{i=1}^N$ where $\pi$ is the permutation that sorts $\{\|h_i^t\|_2\}_{i=1}^N$ in the order statistic tree.

The order statistic tree along with the array storing the cumulative sums allows the retrieval of $\rho_t$ in $O(\log^2 N)$ time. The cumulative sums allow to sample from $p^t$ in $O(\log N)$ time using binary search, and maintaining them is the only operation that requires a vectorized $O(N)$ operation. All other operations run in $O(\log N)$ time. See appendix A for the full version of the algorithm and a more complete discussion of its computational complexity.

## 4 Theory

In this section, we prove a sub-linear dynamic regret guarantee for our proposed algorithm when used with SGD and SGLD with decreasing step-sizes. We present our results in a more general setting, and show that they apply to our cases of interest. We start by stating our assumptions:

**Assumption 1.** *(Bounded gradients) There exists a $G > 0$ such that $\|\nabla f_i(x)\|_2 \leq G$ for all $x \in \mathbb{R}^d$ and for all $i \in [N]$.*

**Assumption 2.** *(Smoothness) There exists an $L > 0$ such that $\|\nabla f_i(x) - \nabla f_i(y)\|_2 \leq L \|x - y\|_2$ for all $x, y \in \mathbb{R}^d$ and for all $i \in [N]$.*

**Assumption 3.** *(Contraction of the iterates in expectation) There exists constants $A \geq 0$, $B \geq 1$ and $\delta \in (0, 1]$ such that $\mathbb{E}\left[\|x_{t+1} - x_t\|_2 \mid I_1, \ldots, I_{t-1}\right] \leq A/(B + t - 1)^\delta$ for all $t \in [T]$.*

The bounded gradients assumption has been used in all previous related work [10, 11, 12, 13], although it is admittedly quite strong. The smoothness assumption is standard in the study of optimization and sampling algorithms. Note that we chose to state our assumptions using index-independent constants to make the presentation clearer and since this does not affect our derivation of the sequence $\{\varepsilon_t\}_{t=1}^T$.

Finally, Assumption 3 should really be derived from more basic assumptions, and is only stated to allow for a unified analysis. Note that in the optimization case this is a very mild assumption since we are generally only interested in convergent sequences, and for any such sequence with reasonably fast convergence this assumption holds. The following proposition shows that Assumption 3 holds for our cases of interest. All the proofs for this section can be found in appendix B.

**Proposition 1.** *For any choice of $\{p^t\}_{t=1}^T$, the iterates of SGD (2) with the gradient estimator (4) and decreasing step-sizes $\alpha_t := E/(F + t - 1)^\beta$ with $E \geq 0$, $F \geq 1$ and $\beta \in (0, 1]$ satisfy Assumption 3*

with $A := NGE$, $B := F$, and $\delta := \beta$. Under the same conditions, the iterates of SGLD (3) satisfy Assumption 3 with $A := \sqrt{E}\left(NG\sqrt{\alpha_1} + \sqrt{2d}\right)$, $B := F$, and $\delta := \beta/2$.

We now state a proposition that relates the optimal function value for the problem (13) over the restricted simplex, with the optimal function value over the whole simplex. Its proof is taken from ([12], Lemma 6):

**Proposition 2.** Let $\{a_i\}_{i=1}^N$ be a non-negative set of numbers, and let $\varepsilon \in [0, 1/2N]$. Then:

$$\min_{p \in \Delta(\varepsilon)} \sum_{i=1}^N \frac{1}{p_i} a_i^2 - \min_{p \in \Delta} \sum_{i=1}^N \frac{1}{p_i} a_i^2 \leq 6\varepsilon N \left(\sum_{i=1}^N a_i\right)^2$$

The following lemma gives our first bound on the regret per time step:

**Lemma 2.** Let $q^t := \mathrm{argmin}_{p \in \Delta}\{c_t(p)\}$. Under Assumption 1, and when using the sequence of distributions defined by (10), we have the following bound for $t \in \{t_0, \ldots, T\}$:

$$\mathbb{E}\left[c_t(p^t) - c_t(q^t)\right] \leq \frac{4G}{\varepsilon_t} \mathbb{E}\left[\sum_{i=1}^N \left\|g_i^t - h_i^t\right\|_2\right] + 6\varepsilon_t G^2 N^3$$

where $t_0 := \min\{t \in [T] \mid \varepsilon_t \leq \frac{1}{2N}\}$.

*Proof outline.* Let $\widetilde{p}^t := \mathrm{argmin}_{p \in \Delta}\{\widetilde{c}_t(p)\}$. Then we have the following decomposition:

$$\mathbb{E}\left[c_t(p^t) - c_t(q^t)\right] = \underbrace{\mathbb{E}\left[c_t(p^t) - \widetilde{c}_t(p^t)\right]}_{(A)} + \underbrace{\mathbb{E}\left[\widetilde{c}_t(p^t) - \widetilde{c}_t(\widetilde{p}^t)\right]}_{(B)} + \underbrace{\mathbb{E}\left[\widetilde{c}_t(\widetilde{p}^t) - c_t(q^t)\right]}_{(C)}$$

The terms (A) and (C) are the penalties we pay for using a surrogate cost function, while (B) is the price we pay for restricting the simplex. Using Assumption 1, proposition 2, and the fact that $p^t$ is contained in the $\varepsilon_t$-restricted simplex, each of these terms can be bound to give the result stated. □

The expectation in the first term of the above lemma is highly reminiscent of the first term of the Lyapunov function used to study the convergence of SAGA first proposed in [30] and subsequently refined in [31] (with $g_i^t$ replaced by $g_i^*$, the gradient at the minimum). Inspired by this similarity, we prove the following recursion:

**Lemma 3.** Under Assumptions 2 and 3, we have:

$$\mathbb{E}\left[\sum_{i=1}^N \left\|g_i^{t+1} - h_i^{t+1}\right\|_2\right] \leq \frac{NLA}{(B+t-1)^\delta} + (1 - \varepsilon_t)\mathbb{E}\left[\sum_{i=1}^N \left\|g_i^t - h_i^t\right\|_2\right]$$

The natural thing to do at this stage is to unroll the above recursion, replace in Lemma 2, sum over all time steps, and minimize the obtained regret bound over the choice of the sequence $\{\varepsilon_t\}_{t=1}^T$. However, even if we can solve the resulting optimization problem efficiently, the solution will still depend on the constants $G$ and $L$ and on the initial error due to the initialization of the $h_i$s, both of which are usually unknown. Here instead, we make an asymptotic argument to find the optimal decay rate of $\{\varepsilon_t\}_{t=1}^T$, propose a sequence that satisfies this decay rate, and show that it gives rise to the dynamic regret bounds stated.

Denote by $\varphi(t)$ the expectation in the first term of the upper bound in Lemma 2. Suppose we take $\varepsilon_t$ to be of order $t^{-\beta}$. Looking at the recursion in Lemma 3, we see that to control the positive term, we need the negative term to be of order at least $t^{-\delta}$, so that $\varphi(t)$ cannot be smaller than $t^{\beta-\delta}$. The bound of Lemma 2 is therefore of order $t^{2\beta-\delta} + t^{-\beta}$. The minimum is attained when the exponents are equal so we have: $2\beta - \delta = -\beta \implies \beta = \frac{\delta}{3}$

We are now ready to guess the form of $\varepsilon_t$. Matching the positive term in Lemma 3, we consider the following sequence:

$$\varepsilon_t := \frac{1}{C^{1-\delta/3}(C + t - 1)^{\delta/3}} \tag{15}$$

For a free parameter $C$ satisfying $C \geq N$ to ensure $\varepsilon_1 \leq 1/N$. With this choice of the sequence $\{\varepsilon_t\}_{t=1}^T$, we are now finally ready to state our main result of the paper:

**Theorem 1.** *Under Assumptions 1 and 2 on the functions $f_i$ and Assumption 3 on the sequence $\{x_t\}_{t=1}^T$, algorithm (10) with the sequence $\{\varepsilon_t\}_{t=1}^T$ given in (15) satisfies the following dynamic regret bound for all $T \geq t_0$:*

$$\mathbb{E}\left[Regret_D(T)\right] \leq \mathcal{O}(T^{1-\delta/3}) \tag{16}$$

*where $t_0 := \min\{t \in [T] \mid \varepsilon_t \leq \frac{1}{2N}\}$ as in Lemma 2.*

*Proof outline.* Using the sequence given by (15) and the recursion in Lemma 3, we show by induction that $\varphi(t) \leq \mathcal{O}(t^{-2\delta/3})$ . Replacing in Lemma 2, summing over all time steps, and bounding the resulting sums by the appropriate integrals we get the result. □

Note that since $C \geq N$, we have $t_0 \leq (2^{3/\delta} - 1)N + 1$, so $t_0$ is bounded by a constant independent of $T$. Furthermore, setting $C = 2N$ makes the theorem hold for all $T \in \mathbb{N}$. In practice however, it might be beneficial to set $C = N$ to overcome a bad initialization of the $h_i$s.

Combining Theorem 1 with proposition 1 we obtain the following corollary:

**Corollary 1.** *Under Assumptions 1 and 2 on the functions $f_i$, if SGD (2) is run with step-sizes $\mathcal{O}(1/t)$ using the estimator (4) and probabilities (10) with the sequence $\{\varepsilon_t\}_{t=1}^T$ given by (15), then for all $T \geq t_0$:*

$$\mathbb{E}\left[Regret_D(T)\right] \leq \mathcal{O}(T^{2/3}) \tag{17}$$

*and for SGLD (3):*

$$\mathbb{E}\left[Regret_D(T)\right] \leq \mathcal{O}(T^{5/6}) \tag{18}$$

*under the same conditions.*

## 5   A new mini-batch estimator

In most practical applications, one uses a mini-batch of samples to construct the gradient estimator instead of just a single sample. The most basic such estimator is the one formed by sampling a mini-batch of indices $S_t = \{I_t^1, \ldots, I_t^m\}$ uniformly and independently, and taking the sample mean. This gives an unbiased estimator, whose variance decreases as $1/m$. A simple way to make it more efficient is by sampling the indices uniformly but without replacement. In that case, the variance decreases by an additional factor of $(1 - (m-1)/(N-1))$. For $m \ll N$, the difference is negligible, and the additional cost of sampling without replacement is not justified. However, when using unequal probabilities, this argument no longer holds, and the additional variance reduction obtained from sampling without replacement can be significant even for small $m$.

For our problem, besides the additional variance reduction, sampling without replacement allows a higher rate of replacement of the $h_i$s, which is directly related to our regret through the factor in front of the second term of Lemma 3, whose proper generalization for mini-batch sampling is $(1 - \min_{i \in [N]} \pi_i^t)$, where $\pi_i^t = P(i \in S_t)$ is the inclusion probability of index $i$. This makes sampling without replacement desirable for our purposes. Unfortunately, unlike in the uniform case, the sample mean generalization of (4) is no longer unbiased. We propose instead the following estimator:

$$\hat{g}_b^t = \frac{1}{m} \sum_{j=1}^m \hat{g}_j^t \qquad \hat{g}_j^t := \left[\frac{1}{q_{I_t^j}^{t,j}} g_{I_t^j}^t + \sum_{k=1}^{j-1} g_{I_t^k}^t\right] \qquad q_i^{t,j} := \frac{p_i^t}{1 - \sum_{k=1}^{j-1} p_{I_t^k}^t} \tag{19}$$

We summarize some of its properties in the following proposition:

**Proposition 3.** *Let $S_t^j := \{I_t^1, \ldots, I_t^j\}$ for $j \in [m]$ and $S_t^0 := \emptyset$. We have:*

(a) $\mathbb{E}\left[\hat{g}_b^t\right] = g^t$

(b) $\mathbb{E}\left[\|\hat{g}_b^t - g^t\|_2^2\right] = (1/m^2) \sum_{j=1}^m \mathbb{E}\left[\|\hat{g}_j^t - g^t\|_2^2\right]$

(c) $\operatorname{argmin}_{p \in \Delta}\{\mathbb{E}\left[\|\hat{g}_b^t - g^t\|_2^2\right]\} = \operatorname{argmin}_{p \in \Delta}\{c_t(p)\}$

(d) $\mathbb{E}\left[\|\hat{g}_{j+1}^t - g^t\|_2^2\right] = \left(1 - \mathbb{E}\left[q_{I_t^j}^{t,j}\right]\right)\mathbb{E}\left[\|\hat{g}_j^t - g^t\|_2^2\right] - \mathbb{E}\left[q_{I_t^j}^{t,j}\|\hat{g}_j^t - g^t\|_2^2\right]$

*where all the expectations in (d) are conditional on $S_t^{j-1}$.*

The proposed estimator is therefore unbiased and its variance decreases super-linearly in $m$ (by (b) and (d)). Although we were unable to prove a regret bound for this estimator, (c) suggests that it is still reasonable to use our algorithm. To take into account the higher rate of replacement of the $h_i$s, we propose using the following $\varepsilon_t$ sequence, which is based on the mini-batch equivalent of Lemma 3 and the inequality $\min_{i \in [N]} \pi_i^t \geq m\varepsilon_t$ [32, 33, 34]:

$$\varepsilon_t := \frac{1}{C^{1-\delta/3}(C + m(t-1))^{\delta/3}} \tag{20}$$

## 6 Experiments

In this section, we present results of experiments with our algorithm. We start by validating our theoretical results through a synthetic experiment. We then show that our proposed method outperforms existing methods on real world datasets. Finally, we identify settings in which adaptive importance sampling can lead to significant performance gains for the final optimization algorithm.

In all experiments, we added $l_2$-regularization to the model's loss and set the regularization parameter $\mu = 1$. We ran SGD (2) with decreasing step sizes $\alpha_t = \frac{m}{2NL+m\mu t}$ where $m$ is the batch size following [35]. We experimented with 3 different samplers in addition to our proposed sampler: Uniform, Multi-armed bandit Sampler (*Mabs*) [11], and Variance Reducer Bandit (*Vrb*) [12]. The hyperparameters of both *Mabs* and *Vrb* are set to the ones prescribed by the theory in the original papers. For our propsoed sampler *Avare*, we use the epsilon sequence given by (20) with $C = N$, $\delta = 1$, and initialized $h_i = 0$ for all $i \in [N]$. For each sampler, we ran SGD 10 times and averaged the results. The shaded areas represent a one standard deviation confidence interval.

To validate our theoretical findings, we randomly generated a dataset for binary classification with $N = 100$ and $d = 10$. We then trained a logistic regression model on the generated data, and used a batch size of 1 to match the setting of our regret bound. The results of this experiment are depicted in figure 1, and show that *Avare* outperforms the other methods, achieving significantly lower dynamic regret and faster convergence.

For our second experiment, we tested our algorithm on three real world datasets: MNIST, IJCNN1 [36], and CIFAR10. We used a softmax regression model, and a batch size of 128 sampled with replacement. The results of this experiment can be seen in figure 2. The third column shows the relative error which we define as $[c_t(p^t) - \min_{p \in \Delta} c_t(p)] / \min_{p \in \Delta} c_t(p)$. In all three cases, *Avare* achieves significantly smaller dynamic regret, with a relative error quickly decaying to zero. The performance gains in the final optimization algorithm are clear in both MNIST and IJCNN1, but are not noticeable in the case of CIFAR10.

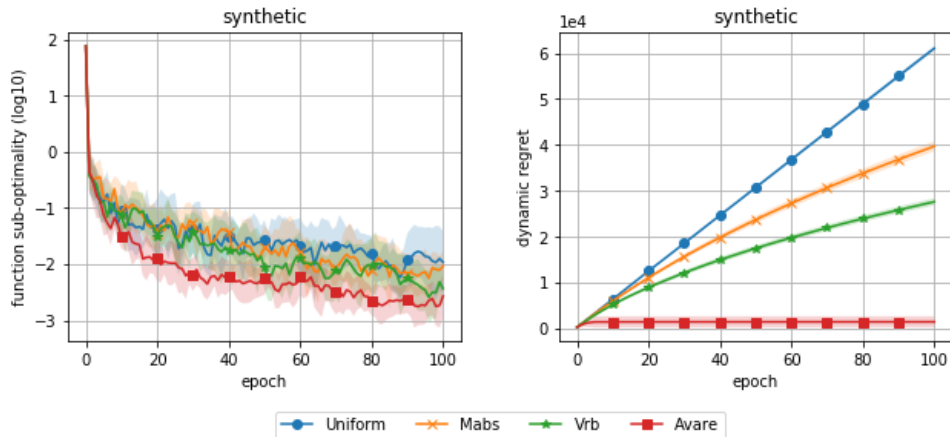

Figure 1: Evolution of function sub-optimality (left) and dynamic regret (right) as a function of data passes on a synthetic dataset with an $l_2$ regularized logistic regression model and different samplers.

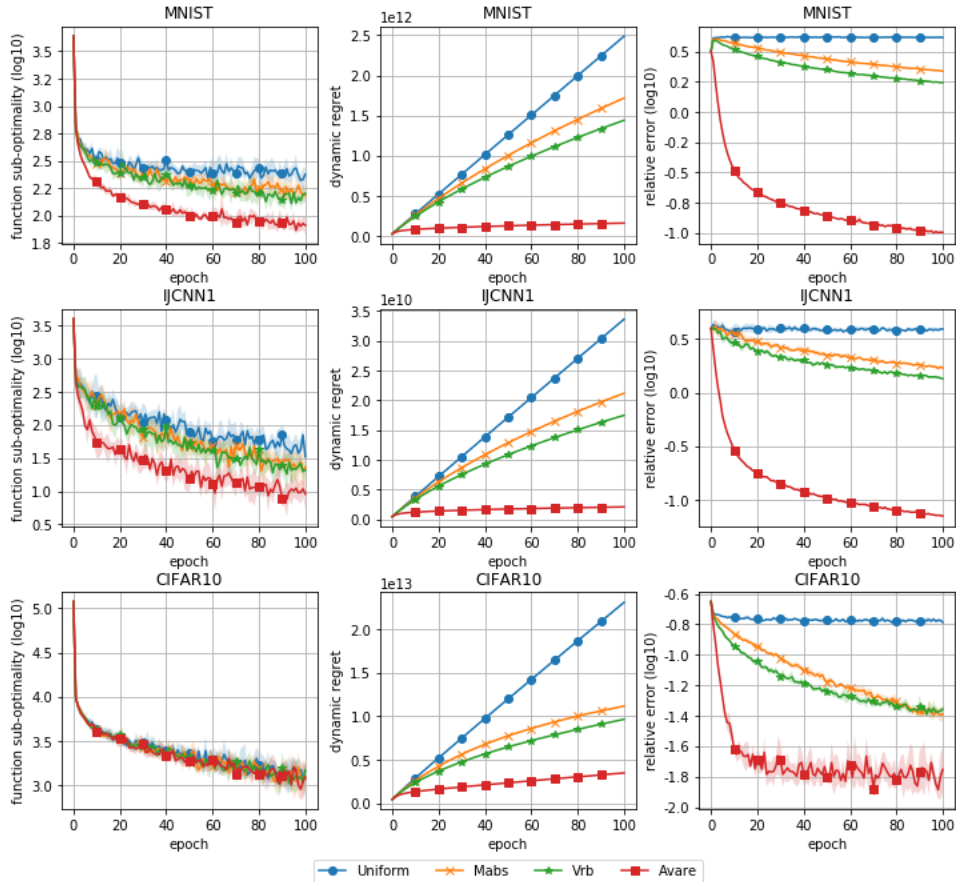

Figure 2: Comparison of the performance of importance samplers on an l2-regularized softmax regression model on three real world datasets: MNIST (top), IJCNN1 (middle), CIFAR10 (bottom).

Table 1: Useful ratios in determining the effectiveness of variance reduction through importance sampling. $L_i$ is the smoothness constant of $f_i$, $L_{max} = \max_{i \in [N]} L_i$, and $g_i^*$ and is the gradient of $f_i$ at the loss minimizer $x^*$.

| Dataset | $\frac{NL_{max}}{\sum_{i=1}^{N} L_i}$ | $\frac{N \sum_{i=1}^{N} \|g_i^*\|_2^2}{(\sum_{i=1}^{N} \|g_i^*\|_2)^2}$ |
|---------|---------|---------|
| Synthetic | 1.69 | 4.46 |
| MNIST | 3.28 | 5.08 |
| IJCNN1 | 1.12 | 4.83 |
| CIFAR10 | 3.40 | 1.14 |

To determine the effectiveness of non-uniform sampling in accelerating the optimization process, previous work [4, 11] has suggested to look at the ratio of the maximum smoothness constant and the average smoothness constant. We show here that this is the wrong measure to look at when using adaptive probabilities. In particular, we argue that the ratio of the variance with uniform sampling at the loss minimizer to the optimal variance at the loss minimizer is much more informative of the performance gains achievable through adaptive importance sampling. For each dataset, these ratios are displayed in Table 1, supporting our claim. We suspect that for large models capable of fitting the data nearly perfectly, our proposed ratio will be large since many of the per-example gradients at the optimum will be zero. We therefore expect our method to be particularly effective in the training of models of this type. We leave such experiments for future work. Finally, in appendix D, we propose an extension of our method to constant step-size SGD, and show that it preserves the performance gains observed when using decreasing step-sizes.

## Broader Impact

Our work develops a new method for variance reduction for stochastic optimization and sampling algorithms. On the optimization side, we expect our method to be very useful in accelerating the training of large scale neural networks, particularly since our method is expected to provide significant performance gains when the model is able to fit the data nearly perfectly. On the sampling side, we expect our method to be useful in accelerating the convergence of MCMC algorithms, opening the door for the use of accurate Bayesian methods at a large scale.

## Acknowledgments and Disclosure of Funding

This research was supported by an NSERC discovery grant. We would like to thank the anonymous reviewers for their useful comments and suggestions.

## Footnotes

[1] Up to a normalizing factor of $\frac{1}{N}$ which does not affect the optimization

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
