[Supplementary Material]

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

*Proof.*

**Edge case and well-definedness.**  If $a_i = 0$ for all $i \in [N]$, then the objective function is identically $0$ over $\Delta(\varepsilon)$ for all $\varepsilon \in [0, 1/N]$, so that any $p \in \Delta(\varepsilon)$ is a solution (we set $(1/0)0 := 0$ in the objective). Else there exists an $i \in [N]$ such that $a_i > 0$, and therefore $a_{\pi(1)} > 0$. Now:

$$\frac{\varepsilon}{1 - (N - 1)\varepsilon} \leq 1$$
$$\Leftrightarrow \varepsilon \leq 1 - (N - 1)\varepsilon$$
$$\Leftrightarrow 0 \leq 1 - N\varepsilon$$
$$\Leftrightarrow N\varepsilon \leq 1$$
$$\Leftrightarrow \varepsilon \leq \frac{1}{N}$$

The last inequality is true, so it implies the first, and we have:

$$a_{\pi(1)} \geq \varepsilon \frac{a_{\pi(1)}}{1 - (N - 1)\varepsilon}$$

Therefore $\rho$ is well defined and is $\geq 1$. As a consequence, $\lambda$ is also well defined and is $> 0$, making $p^*$ in turn well-defined.

**Optimality proof.**  It is easily verified that problem (13) is convex. Its Lagrangian is given by:

$$\mathcal{L}(p, \mu, \nu) = \sum_{i=1}^{N} \frac{1}{p_i} a_i^2 - \sum_{i=1}^{N} \mu_i (p_i - \varepsilon) + \nu \left( \sum_{i=1}^{N} p_i - 1 \right)$$

and the KKT conditions are:

$$\text{(Stationarity)} \quad p_i = \frac{a_i}{\sqrt{\nu - \mu_i}}$$

$$\text{(Complementary slackness)} \quad \mu_i = 0 \vee p_i = \varepsilon$$

$$\text{(Primal feasibility)} \quad p_i \geq \varepsilon \wedge \sum_{j=1}^{N} p_j = 1$$

$$\text{(Dual feasibility)} \quad \mu_i \geq 0$$

By convexity of the problem, the KKT conditions are sufficient conditions for global optimality. To show that our proposed solution is optimal, it therefore suffices to exhibit constants $(\mu_i^*)_{i=1}^N$ and $\nu^*$ that together with $p^*$ satisfy these conditions. Let:

$$\nu^* := \lambda^2$$

$$\mu_i^* := \begin{cases} 0 & \text{if } i \in \{\pi(1), \dots, \pi(\rho)\} \\ \nu^* - a_i^2/\varepsilon^2 & \text{otherwise} \end{cases}$$

Note that the $\mu_i^*$s are well defined since when $\varepsilon = 0$, $\rho = N$, and $\mu_i^* = 0$ for all $i \in [N]$. We claim that the triplet $(p^*, (\mu_i^*)_{i=1}^N, \nu^*)$ satisfies the KKT conditions.

Stationarity and complementary slackness are immediate from the definitions. The first clause of primal feasibility holds by definition of $p_i^*$ for $i \in \{\pi(\rho+1), \dots, \pi(N)\}$. In the other case we have:

$$p_i^* = \frac{a_i}{\lambda}$$

$$= \frac{a_i}{\sum_{j=1}^\rho a_{\pi(j)}} \left(1 - (N - \rho)\varepsilon\right)$$

$$\geq \frac{a_{\pi(\rho)}}{\sum_{j=1}^\rho a_{\pi(j)}} \left(1 - (N - \rho)\varepsilon\right)$$

$$\geq \varepsilon$$

where in the third line we used the fact that $\pi$ orders $\{a_i\}_{i=1}^N$ in decreasing order, and in the last line we used the inequality in the definition of $\rho$. For the second clause of primal feasibility:

$$\sum_{j=1}^N p_j^* = \sum_{j=1}^N p_{\pi(j)}^* = \sum_{j=1}^\rho \frac{a_{\pi(i)}}{\lambda} + \sum_{j=\rho+1}^N \varepsilon = (1 - (N - \rho)\varepsilon) + (N - \rho)\varepsilon = 1$$

Finally, dual feasibility holds by definition of $\mu_i^*$ for $i \in \{\pi(1), \dots, \pi(\rho)\}$. In the other case we have:

$$\mu_i^* = \nu^* - \frac{a_i^2}{\varepsilon^2}$$

$$= \lambda^2 - \frac{a_i^2}{\varepsilon^2}$$

$$= \left(\lambda + \frac{a_i}{\varepsilon}\right)\left(\lambda - \frac{a_i}{\varepsilon}\right)$$

$$\geq \left(\lambda + \frac{a_i}{\varepsilon}\right)\left(\lambda - \frac{a_{\pi(\rho+1)}}{\varepsilon}\right)$$

The first factor is positive by positivity of $\lambda$ and non-negativity of the $a_i$s. For the second factor, we have by the maximality of $\rho$:

$$a_{\pi(\rho+1)} < \varepsilon \frac{\sum_{j=1}^{\rho+1} a_{\pi(j)}}{1 - (N - \rho - 1)\varepsilon}$$

$$\Rightarrow a_{\pi(\rho+1)}(1 - (N - \rho)\varepsilon + \varepsilon) < \varepsilon \sum_{i=1}^\rho a_{\pi(j)} + \varepsilon a_{\pi(\rho+1)}$$

$$\Rightarrow a_{\pi(\rho+1)}(1 - (N - \rho)\varepsilon) < \varepsilon \sum_{i=1}^\rho a_{\pi(j)}$$

$$\Rightarrow \frac{a_{\pi(\rho+1)}}{\varepsilon} < \lambda \tag{21}$$

Therefore the second factor is also positive, and dual feasibility holds. $\qquad\square$

## A.2 Implementation and complexity

In this section of the appendix, we provide pseudocode for the implementation of the algorithm and discuss its computational complexity.

**Algorithm 1** Implementation of the proposed sampler

1: **class** SAMPLER:
2:     **procedure** INITIALIZE($\{\|h_i\|_2\}_{i=1}^N$)
3:         $self.H \leftarrow Array(\{\|h_i\|_2\}_{i=1}^N)$
4:         $self.T \leftarrow OST(keys = \{\|h_i\|_2\}_{i=1}^N, values = [N])$
5:         $self.CS \leftarrow Array(\{\sum_{j=1}^i \|h_{\pi(j)}\|_2\}_{i=1}^N)$

6:     **procedure** DELETE($x$)
7:         $r \leftarrow self.T.rank(x)$
8:         $self.T.delete(x)$
9:         $self.CS[r:N] \leftarrow self.CS[r:N] - x$

10:     **procedure** INSERT($x$, $i$)
11:         $self.T.insert(key = x, value = i)$
12:         $r \leftarrow self.T.rank(x)$
13:         $self.CS[r:N] \leftarrow self.CS[r:N] + x$

14:     **procedure** UPDATE($\|h_I\|_2$, $I$)
15:         $self.delete(self.H[I])$
16:         $self.H[I] \leftarrow \|h_I\|_2$
17:         $self.insert(\|h_I\|_2, I)$

18:     **function** SEARCH($\varepsilon$, node)
19:         $r \leftarrow self.T.rank(node)$
20:         **if** $r == N$ **then return** $r$
21:         $c \leftarrow 1 - (N - r)\varepsilon$
22:         **if** $c \cdot node.key < \varepsilon \cdot self.CS[r]$ **then**
23:             **return** $self.search(\varepsilon, node.left)$
24:         **else**
25:             $d = 1 - (N - r - 1)\varepsilon$
26:             **if** $d \cdot node.successor.key < \varepsilon \cdot self.CS[r + 1]$ **then**
27:                 **return** $r$
28:             **else**
29:                 **return** $self.search(\varepsilon, node.right)$

30:     **function** SAMPLE($\varepsilon$)
31:         $\rho \leftarrow self.search(\varepsilon, self.T.root)$
32:         $\lambda \leftarrow self.CS[\rho] / (1 - (N - \rho)\varepsilon)$
33:         $b \sim Bernoulli((N - \rho)\varepsilon)$
34:         **if** b == 1 **then**
35:             Sample an index $\tilde{I}$ uniformly from $\{\rho + 1, \ldots, N\}$.
36:         **else**
37:             $u \sim Uniform([0, 1])$
38:             Find the first index $\tilde{I}$ for which $\lambda u \leq self.CS[\tilde{I}]$ using binary search.
39:         $I \leftarrow self.T.select(\tilde{I})$
40:         **return** $I$
41:

---
**Algorithm 2** AVARE
---
    **Input**: $x_1, T, \{\alpha_t\}_{t=1}^T, \{\|h_i^1\|_2\}_{i=1}^N, C \geq N$

1:   $sampler \leftarrow \text{SAMPLER}(\{\|h_i^1\|_2\}_{i=1}^N)$

2:   **for** $t = 1, 2, \ldots, T$ **do**

3:        Set $\varepsilon_t$ according to (15)

4:        $I_t \leftarrow sampler.sample(\varepsilon_t)$

5:        Obtain $x_{t+1}$ using (2) or (3) and the estimator (4).

6:        $sampler.update(\|g_{I_t}\|_2, I_t)$

7:   **return** $x_{T+1}$
---

### A.2.1 Implementation

We assume in algorithm 1 that an order statistic tree (OST) (see, for example, [38], chapter 14) can be instantiated and that the ordering in the tree is such that the key of the left child of a node is greater than or equal to that of the node itself. Furthermore, we assume that the $rank(x)$ method returns the position of $x$ in the order determined by an inorder traversal of the tree. Finally, the $select(i)$ method returns the value of the $i^{th}$-largest key in the tree.

The algorithm works as follows. At initialization, three data structures are initialized: an arrray $H$ holding the gradient norms according to the original indices, an order statistic tree $T$ holding the gradient norms as keys and the original indices as values, and an array $CS$ holding the cumulative sums of the gradient norms, where the sums are accumulated in the (decreasing) order that sorts the gradients norms in $T$.

The $sample(\varepsilon)$ method allows to sample from the optimal distribution on $\Delta(\varepsilon)$. It uses the $search(\varepsilon, node)$ method to find $\rho$ by searching the tree $T$ and using the maximality property of $\rho$. Once $\rho$ is determined, $\lambda$ can be calculated. Using the fact that the cumulative sums are proportional to the CDF of the distribution, the algorithm then samples an index using inverse-transform sampling. The sampled index is then transformed back to an index in the original order using the $select$ method of the tree $T$.

Finally, the $update(\|h_I\|_2, I)$ method replaces the gradient norm of a given index by a new one. It calls the methods $delete(x)$ and $insert(x, i)$ which perform the deletion and insertion while maintaining the tree $T$ and array $CS$.

### A.2.2 Complexity

First, let us analyze the cost of running the $update(\|h_I\|_2, I)$ method. For the array $H$, we only use random access and assignment, which are both $O(1)$. For the tree $T$, we use the methods $insert$, $delete$, and $rank$, all of which are $O(\log N)$. Finally, for the array $CS$ we add and subtract from a sub-array, which takes $O(N)$ time, although this operation is vectorized and very fast in practice.

Let us now look at the cost of running $sample(\varepsilon)$. The $search$ method is recursive, but will only be called at most as many times as the height of the tree, which is $O(\log N)$. Now for each call of $search$, both the $rank$ and $successor$ methods of the tree $T$ require $O(\log N)$ time. The rest of the $search$ method only requires $O(1)$ operations. The total cost of the $search$ method is therefore $O(\log^2 N)$. For the rest of the $sample$ method, the operations that dominate the cost are the $select$ method of the tree $T$, which takes $O(\log N)$ time, and the binary search in the else branch, which also runs in $O(\log N)$ time. Consequently, the total cost of the sample method is $O(\log^2 N)$.

The total per iteration cost of using the proposed sampler is therefore $O(N)$ vectorized operations, and $O(\log^2 N)$ sequential (non-vectorized) operations. The total memory cost is $O(N)$.

## Appendix B  Theory

We restate our assumptions here for ease of reference.

**Assumption 1.** *(Bounded gradients) There exists a $G > 0$ such that $\|\nabla f_i(x)\|_2 \leq G$ for all $x \in \mathbb{R}^d$ and for all $i \in [N]$.*

**Assumption 2.** *(Smoothness) There exists an $L > 0$ such that $\|\nabla f_i(x) - \nabla f_i(y)\|_2 \leq L \|x - y\|_2$ for all $x, y \in \mathbb{R}^d$ and for all $i \in [N]$.*

**Assumption 3.** *(Contraction of the iterates in expectation) There exists constants $A \geq 0$, $B \geq 1$ and $\delta \in (0, 1]$ such that $\mathbb{E}\left[\|x_{t+1} - x_t\|_2 \mid I_1, \ldots, I_{t-1}\right] \leq A/(B + t - 1)^\delta$ for all $t \in [T]$.*

## B.1 Proof of proposition 1

**Proposition 1.** *For any choice of $\{p^t\}_{t=1}^T$, the iterates of SGD (2) with the gradient estimator (4) and decreasing step-sizes $\alpha_t := E/(F + t - 1)^\beta$ with $E \geq 0$, $F \geq 1$ and $\beta \in (0, 1]$ satisfy Assumption 3 with $A := NGE$, $B := F$, and $\delta := \beta$. Under the same conditions, the iterates of SGLD (3) satisfy Assumption 3 with $A := \sqrt{E}\left(NG\sqrt{\alpha_1} + \sqrt{2d}\right)$, $B := F$, and $\delta := \beta/2$.*

*Proof.* Conditioning on the knowledge of $\{I_1, \ldots, I_{t-1}\}$ we have for SGD:

$$\mathbb{E}\left[\left\|x_{t+1}^{SGD} - x_t^{SGD}\right\|_2\right] = \mathbb{E}\left[\alpha_t \frac{1}{p_{I_t}^t}\left\|g_{I_t}^t\right\|_2\right]$$

$$= \alpha_t \sum_{i=1}^N \left\|g_i^t\right\|_2$$

$$\leq \alpha_t NG$$

and for SGLD we have:

$$\mathbb{E}\left[\left\|x_{t+1}^{SGLD} - x_t^{SGLD}\right\|_2\right] \leq \mathbb{E}\left[\alpha_t \frac{1}{p_{I_t}^t}\left\|g_{I_t}^t\right\|_2\right] + \mathbb{E}\left[\|\xi_t\|_2\right]$$

$$\leq \alpha_t \sum_{i=1}^N \left\|g_i^t\right\|_2 + \sqrt{\mathbb{E}\left[\|\xi_t\|_2^2\right]}$$

$$\leq \alpha_t NG + \sqrt{\alpha_t}\sqrt{2d}$$

$$\leq \sqrt{\alpha_t}\left(NG\sqrt{\alpha_1} + \sqrt{2d}\right)$$

where in the first line we used the triangle inequality, in the second we used Jensen's inequality, and in the last we used the fact that $\{\alpha_t\}_{t=1}^T$ is decreasing. Replacing with the value of $\alpha_t$ we obtain the result. $\square$

## B.2 Proof of proposition 2

The following proof is taken from ([12], Lemma 6).

**Proposition 2.** *Let $\{a_i\}_{i=1}^N$ be a non-negative set of numbers, and let $\varepsilon \in [0, 1/2N]$. Then:*

$$\min_{p \in \Delta(\varepsilon)} \sum_{i=1}^N \frac{1}{p_i}a_i^2 - \min_{p \in \Delta} \sum_{i=1}^N \frac{1}{p_i}a_i^2 \leq 6\varepsilon N \left(\sum_{i=1}^N a_i\right)^2$$

*Proof.* By Lemma 1 we have:

$$\min_{p \in \Delta(\varepsilon)} \sum_{i=1}^N \frac{1}{p_i}a_i^2 = \lambda \sum_{i=1}^\rho a_{\pi(i)} + \sum_{i=\rho+1}^N \frac{a_{\pi(i)}^2}{\varepsilon}$$

$$\leq \lambda^2 \left(1 - (N - \rho)\varepsilon\right) + \varepsilon \sum_{i=\rho+1}^N \frac{a_{\pi(\rho+1)}^2}{\varepsilon^2}$$

$$\leq \lambda^2 \left(1 - (N - \rho)\varepsilon\right) + (N - \rho)\varepsilon\lambda^2$$

$$= \lambda^2$$

where in the third line we used inequality (21) from the proof of Lemma 1. Now for the case $\varepsilon = 0$ we have $\rho = N$, so the second term in the first line is zero and the inequality becomes an equality:

$$\min_{p \in \Delta} \sum_{i=1}^{N} \frac{1}{p_i} a_i^2 = \left( \sum_{i=1}^{N} a_i \right)^2 \tag{22}$$

The difference is therefore bounded by:

$$\min_{p \in \Delta(\varepsilon)} \sum_{i=1}^{N} \frac{1}{p_i} a_i^2 - \min_{p \in \Delta} \sum_{i=1}^{N} \frac{1}{p_i} a_i^2 \leq \frac{\left( \sum_{i=1}^{\rho} a_{\pi(i)} \right)^2}{(1 - (N - \rho)\varepsilon)^2} - \left( \sum_{i=1}^{N} a_i \right)^2$$

$$\leq \left( \frac{1}{(1 - N\varepsilon)^2} - 1 \right) \left( \sum_{i=1}^{N} a_i \right)^2$$

$$\leq 6\varepsilon N \left( \sum_{i=1}^{N} a_i \right)^2$$

where in the last line we used the inequality $\frac{1}{(1-x)^2} - 1 \leq 6x$ for $x \in [0, 1/2]$ which gives the restriction $\varepsilon \in [0, 1/2N]$. $\qquad\square$

## B.3 Proof of Lemma 2

**Lemma 2.** *Let* $q^t := \operatorname{argmin}_{p \in \Delta} \{c_t(p)\}$. *Under Assumption 1, and when using the sequence of distributions defined by (10), we have the following bound for* $t \in \{t_0, \ldots, T\}$:

$$\mathbb{E}\left[ c_t(p^t) - c_t(q^t) \right] \leq \frac{4G}{\varepsilon_t} \mathbb{E}\left[ \sum_{i=1}^{N} \left\| g_i^t - h_i^t \right\|_2 \right] + 6\varepsilon_t G^2 N^3$$

*where* $t_0 := \min\{t \in [T] \mid \varepsilon_t \leq \frac{1}{2N}\}$.

*Proof.* Let $t \geq t_0$ and $\widetilde{p}^t := \operatorname{argmin}_{p \in \Delta} \{\widetilde{c}_t(p)\}$. We have the following decomposition:

$$\mathbb{E}\left[ c_t(p^t) - c_t(q^t) \right] = \underbrace{\mathbb{E}\left[ c_t(p^t) - \widetilde{c}_t(p^t) \right]}_{(A)} + \underbrace{\mathbb{E}\left[ \widetilde{c}_t(p^t) - \widetilde{c}_t(\widetilde{p}^t) \right]}_{(B)} + \underbrace{\mathbb{E}\left[ \widetilde{c}_t(\widetilde{p}^t) - c_t(q^t) \right]}_{(C)}$$

We bound each term separately:

$$(A) = \mathbb{E}\left[ \sum_{i=1}^{N} \frac{1}{p_i^t} \left( \left\| g_i^t \right\|_2^2 - \left\| h_i^t \right\|_2^2 \right) \right]$$

$$= \mathbb{E}\left[ \sum_{i=1}^{N} \frac{1}{p_i^t} \left( \left\| g_i^t \right\|_2 - \left\| h_i^t \right\|_2 \right) \left( \left\| g_i^t \right\|_2 + \left\| h_i^t \right\|_2 \right) \right]$$

$$\leq \mathbb{E}\left[ \sum_{i=1}^{N} \frac{2G}{p_i^t} \left\| g_i^t - h_i^t \right\|_2 \right]$$

$$\leq \frac{2G}{\varepsilon_t} \mathbb{E}\left[ \sum_{i=1}^{N} \left\| g_i^t - h_i^t \right\|_2 \right]$$

where in the third line we used Assumption 1 and the reverse triangle inequality, and in the last line we used the fact that $p^t \in \Delta(\varepsilon_t)$. Since $t \geq t_0$, we can apply proposition 2 on (B) to obtain:

$$(B) \leq \mathbb{E}\left[ 6\varepsilon_t N \left( \sum_{i=1}^{N} \left\| h_i^t \right\|_2 \right)^2 \right] \leq 6\varepsilon_t G^2 N^3$$

where the second inequality uses Assumption 1. Finally, using the optimal function values (22) we have for (C):

$$
\begin{aligned}
(\text{C}) &= \mathbb{E}\left[\left(\sum_{i=1}^{N}\left\|h_i^t\right\|_2\right)^2 - \left(\sum_{i=1}^{N}\left\|g_i^t\right\|_2\right)^2\right] \\
&= \mathbb{E}\left[\left(\sum_{i=1}^{N}\left(\left\|h_i^t\right\|_2 - \left\|g_i^t\right\|_2\right)\right)\left(\sum_{i=1}^{N}(\left\|g_i^t\right\|_2 + \left\|h_i^t\right\|_2)\right)\right] \\
&\leq 2GN\,\mathbb{E}\left[\sum_{i=1}^{N}\left\|g_i^t - h_i^t\right\|_2\right] \\
&\leq \frac{2G}{\varepsilon_t}\,\mathbb{E}\left[\sum_{i=1}^{N}\left\|g_i^t - h_i^t\right\|_2\right]
\end{aligned}
$$

where we again used Assumption 1 and the reverse triangle inequality in the third line. the last inequality follows from the fact that $\varepsilon_t \leq \frac{1}{N}$. Combining the three bounds gives the result. $\qquad\square$

## B.4  Proof of Lemma 3

**Lemma 3.** *Under Assumptions 2 and 3, we have:*

$$
\mathbb{E}\left[\sum_{i=1}^{N}\left\|g_i^{t+1} - h_i^{t+1}\right\|_2\right] \leq \frac{NLA}{(B+t-1)^\delta} + (1-\varepsilon_t)\,\mathbb{E}\left[\sum_{i=1}^{N}\left\|g_i^t - h_i^t\right\|_2\right]
$$

*Proof.* Conditioning on the knowledge of $\{I_1, \ldots, I_{t-1}\}$ we have:

$$
\begin{aligned}
\mathbb{E}\left[\sum_{i=1}^{N}\left\|g_i^{t+1} - h_i^{t+1}\right\|_2\right] &= \sum_{j=1}^{N}p_j^t\left[\left\|g_j^{t+1} - g_j^t\right\|_2 + \sum_{\substack{i=1\\i\neq j}}^{N}\left\|g_i^{t+1} - h_i^t\right\|_2\right] \\
&\leq \sum_{j=1}^{N}p_j^t\left[\left\|g_j^{t+1} - g_j^t\right\|_2 + \sum_{\substack{i=1\\i\neq j}}^{N}\left(\left\|g_i^{t+1} - g_i^t\right\|_2 + \left\|g_i^t - h_i^t\right\|_2\right)\right] \\
&= \sum_{j=1}^{N}p_j^t\left[\sum_{i=1}^{N}\left\|g_i^{t+1} - g_i^t\right\|_2 + \sum_{\substack{i=1\\i\neq j}}^{N}\left\|g_i^t - h_i^t\right\|_2\right] \\
&\leq NL\sum_{j=1}^{N}p_j^t\left\|x_{t+1} - x_t\right\|_2 + \sum_{i=1}^{N}\left(\sum_{\substack{j=1\\j\neq i}}^{N}p_j^t\right)\left\|g_i^t - h_i^t\right\|_2 \\
&= NL\,\mathbb{E}\left[\left\|x_{t+1} - x_t\right\|_2\right] + \sum_{i=1}^{N}(1 - p_i^t)\left\|g_i^t - h_i^t\right\|_2 \\
&\leq \frac{NLA}{(B+t-1)^\delta} + (1-\varepsilon_t)\sum_{i=1}^{N}\left\|g_i^t - h_i^t\right\|_2
\end{aligned}
$$

where in the fourth line we used Assumption 2, and in the last line we used Assumption 3 and the fact that $p^t \in \Delta(\varepsilon_t)$. Taking expectation with respect to the choice of $\{I_1, \ldots, I_{t-1}\}$ on both sides we get the result. $\qquad\square$

## B.5  Lemma 4

Before proving Theorem 1, we first state and prove the following solution of the recursion of Lemma 3 assuming the sequence $\{\varepsilon_t\}_{t=1}^{T}$ is given by (15).

**Lemma 4.** *Assuming the use of the sequence $\{\varepsilon_t\}_{t=1}^T$ given by (15) we have:*

$$\mathbb{E}\left[\sum_{i=1}^N \|g_i^t - h_i^t\|_2\right] \leq \frac{K}{(C+t-1)^{2\delta/3}}$$

*where:*

$$K := \max\left\{\frac{3C^{1-\delta/3}D}{3-2\delta}, C^{2\delta/3}\sum_{i=1}^N \|g_i^1 - h_i^1\|_2\right\}$$

*and:*

$$D := \begin{cases} NLA & \text{if } B \geq C \\ (\frac{C}{B})^\delta NLA & \text{if } B < C \end{cases}$$

*where A, B, and $\delta$ are as in Assumption 3.*

*Proof.*

**A simple inequality.** Suppose $B \geq C$, then:

$$\frac{NLA}{(B+t-1)^\delta} \leq \frac{NLA}{(C+t-1)^\delta}$$

otherwise, we have:

$$\frac{NLA}{(B+t-1)^\delta} = \frac{(\frac{C}{B})^\delta NLA}{(C+(\frac{C}{B})(t-1))^\delta} \leq \frac{(\frac{C}{B})^\delta NLA}{(C+t-1)^\delta}$$

where the last inequality follows from the fact that $C > B$ and $t \geq 1$. We conclude that:

$$\frac{NLA}{(B+t-1)^\delta} \leq \frac{D}{(C+t-1)^\delta}$$

**Induction proof.** Let $\varphi(t) := \mathbb{E}\left[\sum_{i=1}^N \|g_i^t - h_i^t\|_2\right]$, and let $K' := C^{2-2\delta/3}K$. For $t = 1$ the statement holds since:

$$\varphi(1) = \frac{C^{2\delta/3}\varphi(1)}{(C+t-1)^{2\delta/3}} \leq \frac{K}{(C+t-1)^{2\delta/3}}$$

For $t \geq 1$ we have by Lemma 3 and the above inequality:

$$\varphi(t+1) \leq \frac{D}{(C+t-1)^\delta} + \left(1 - \frac{1}{C^{1-\delta/3}(C+t-1)^{\delta/3}}\right)\varphi(t)$$

$$= \frac{aC^{3-\delta}D}{aC^{3-\delta}(C+t-1)^\delta} + \left(1 - \frac{1}{C^{1-\delta/3}(C+t-1)^{\delta/3}}\right)\varphi(t)$$

$$\leq \frac{K'}{aC^{3-\delta}(C+t-1)^\delta} + \left(1 - \frac{1}{C^{1-\delta/3}(C+t-1)^{\delta/3}}\right)\frac{K'}{C^{2-2\delta/3}(C+t-1)^{2\delta/3}}$$

where $a = \frac{3}{3-2\delta}$ and where the last line follows by the induction hypothesis. To simplify notation let $x := (C+t-1)$, $E := C^{1-\delta/3}$, $\gamma := (1-\frac{1}{a}) = \frac{2\delta}{3}$. Then the above inequality can be rewritten as:

$$\varphi(t+1) \leq K'\left(\frac{1}{E^2 x^{2\delta/3}} - \frac{\gamma}{E^3 x^\delta}\right)$$

$$= K'\frac{Ex^{\delta/3} - \gamma}{E^3 x^\delta}$$

$$= K'\frac{E^3 x^\delta - \gamma^3}{E^3 x^\delta(E^2 x^{2\delta/3} + E\gamma x^{\delta/3} + \gamma^2)}$$

$$\leq K'\frac{1}{E^2 x^{2\delta/3} + E\gamma x^{\delta/3}}$$

Now by concavity of $x^{2\delta/3}$ we have:

$$E^2 \left[ (x+1)^{2\delta/3} - x^{2\delta/3} \right] \leq E^2 \frac{2\delta}{3} x^{2\delta/3-1}$$

so that:

$$
\begin{aligned}
&E^2 x^{2\delta/3} + E\gamma x^{\delta/3} \geq E^2 (x+1)^{2\delta/3} \\
&\Leftrightarrow E\gamma x^{\delta/3} \geq E^2 \left[ (x+1)^{2\delta/3} - x^{2\delta/3} \right] \\
&\Leftarrow \gamma E x^{\delta/3} \geq \frac{2\delta}{3} E^2 x^{2\delta/3-1} \\
&\Leftrightarrow x^{1-\delta/3} \geq C^{1-\delta/3} \\
&\Leftrightarrow x \geq C \\
&\Leftrightarrow (C+t-1) \geq C \\
&\Leftrightarrow t \geq 1
\end{aligned}
$$

The last statement is true, and therefore so is the first. Replacing in the bound on $\varphi(t+1)$ we get:

$$\varphi(t+1) \leq \frac{K'}{C^{2-2\delta/3}(C+(t+1)-1)^{2\delta/3}} = \frac{K}{(C+(t+1)-1)^{2\delta/3}}$$

which finishes the proof. □

## B.6 Proof of Theorem 1

**Theorem 1.** *Under Assumptions 1 and 2 on the functions $f_i$ and Assumption 3 on the sequence $\{x_t\}_{t=1}^T$, algorithm (10) with the sequence $\{\varepsilon_t\}_{t=1}^T$ given in (15) satisfies the following dynamic regret bound for all $T \geq t_0$:*

$$\mathbb{E}\left[Regret_D(T)\right] \leq \mathcal{O}(T^{1-\delta/3}) \tag{16}$$

*where $t_0 := \min\{t \in [T] \mid \varepsilon_t \leq \frac{1}{2N}\}$ as in Lemma 2.*

*Proof.* Combining Lemma 4 with Lemma 2 we have the following per-step bound for $t \geq t_0$:

$$\mathbb{E}\left[c_t(p^t) - c_t(q^t)\right] \leq \frac{4GKC^{1-\delta/3} + \frac{6G^2 N^3}{C^{1-\delta/3}}}{(C+t-1)^{\delta/3}} =: \frac{K'}{(C+t-1)^{\delta/3}}$$

Summing over the time steps $\{t_0, \dots, T\}$ we get:

$$\sum_{t=t_0}^T \mathbb{E}\left[c_t(p^t) - c_t(q^t)\right] \leq \sum_{t=t_0}^T \frac{K'}{(C+t-1)^{\delta/3}}$$

Therefore:

$$
\begin{aligned}
\mathbb{E}\left[\text{Regret}_D(T)\right] &= \sum_{t=1}^{t_0-1} \mathbb{E}\left[c_t(p^t) - c_t(q^t)\right] + \sum_{t=t_0}^T \mathbb{E}\left[c_t(p^t) - c_t(q^t)\right] \\
&\leq \sum_{t=1}^{t_0-1} \mathbb{E}\left[c_t(p^t)\right] + \sum_{t=t_0}^T \frac{K'}{(C+t-1)^{\delta/3}} \\
&\leq \sum_{t=1}^{t_0-1} \sum_{i=1}^N \mathbb{E}\left[\frac{1}{p_i^t} \left\|g_i^t\right\|_2^2\right] + K' \int_{t=t_0-1}^T \frac{1}{(C+t-1)^{\delta/3}} dt \\
&\leq (t_0-1)\frac{NG^2}{\varepsilon_{t_0}} + K' \frac{(C+T-1)^{1-\delta/3} - (C+t_0-2)^{1-\delta/3}}{1-\delta/3} \\
&\leq (2^{3/\delta}-1)\frac{N^2 G^2}{\varepsilon_{t_0}} + K' \frac{(C+T-1)^{1-\delta/3} - (C-1)^{1-\delta/3}}{1-\delta/3} \\
&= \mathcal{O}(T^{1-\delta/3})
\end{aligned}
$$

Where the line before the last follows from the fact that $1 \leq t_0 \leq (2^{3/\delta}-1)N+1$ since $C \geq N$. □

## Appendix C   A new mini-batch estimator

### C.1   A class of unbiased estimators

It will be useful for our discussion to consider the following class $\mathcal{C}(p^{t,1}, \ldots, p^{t,m})$ of estimators:

$$\hat{g}_b^t(p^{t,1}, \ldots, p^{t,m}) := \frac{1}{m} \sum_{j=1}^{m} \hat{g}_j^t(p^{t,j}) \qquad \hat{g}_j^t(p^{t,j}) := \left[ \frac{1}{p_{I_t^j}^{t,j}} g_{I_t^j}^t + \sum_{k=1}^{j-1} g_{I_t^k}^t \right]$$

where each $p^{t,j}$ is a distribution on $[N] \setminus \{I_t^1, \ldots, I_t^{j-1}\}$. The estimator we proposed in Section 5 is:

$$\hat{g}_b^t = \frac{1}{m} \sum_{j=1}^{m} \hat{g}_j^t \qquad \hat{g}_j^t := \left[ \frac{1}{q_{I_t^j}^{t,j}} g_{I_t^j}^t + \sum_{k=1}^{j-1} g_{I_t^k}^t \right] \qquad q_i^{t,j} := \frac{p_i^t}{1 - \sum_{k=1}^{j-1} p_{I_t^k}^t} \tag{19}$$

where the indices $S_t = \{I_t^1, \ldots, I_t^m\}$ are sampled without replacement according to $p^t$. Setting:

$$p_i^{t,j} := \begin{cases} 0 & \text{if } i \in \{I_t^1, \ldots, I_t^{j-1}\} \\ q_i^{t,j} & \text{otherwise} \end{cases}$$

we see that our proposed estimator belongs to the class of estimators $\mathcal{C}$ introduced above. The proofs of (a) and (b) of proposition 3 below apply with almost no modification to any estimator in the class $\mathcal{C}$. A natural question then is which estimator in the above-defined class achieves minimum variance. We answer this in the proof of part $(c)$ below, and show that our proposed estimator (19) with $p^t := \operatorname{argmin}_{p \in \Delta} \{c_t(p)\}$ achieves minimum variance.

### C.2   Proof of proposition 3

**Proposition 3.** *Let* $S_t^j := \{I_t^1, \ldots, I_t^j\}$ *for* $j \in [m]$ *and* $S_t^0 := \emptyset$. *We have:*

(a) $\mathbb{E}[\hat{g}_b^t] = g^t$

(b) $\mathbb{E}\left[ \|\hat{g}_b^t - g^t\|_2^2 \right] = (1/m^2) \sum_{j=1}^{m} \mathbb{E}\left[ \|\hat{g}_j^t - g^t\|_2^2 \right]$

(c) $\operatorname{argmin}_{p \in \Delta} \{\mathbb{E}[ \|\hat{g}_b^t - g^t\|_2^2 ]\} = \operatorname{argmin}_{p \in \Delta} \{c_t(p)\}$

(d) $\mathbb{E}\left[ \|\hat{g}_{j+1}^t - g^t\|_2^2 \right] = \left( 1 - \mathbb{E}\left[ q_{I_t^j}^{t,j} \right] \right) \mathbb{E}\left[ \|\hat{g}_j^t - g^t\|_2^2 \right] - \mathbb{E}\left[ q_{I_t^j}^{t,j} \|\hat{g}_j^t - g^t\|_2^2 \right]$

*where all the expectations in (d) are conditional on* $S_t^{j-1}$.

*Proof.*

(a) For $j \in [m]$ and conditional on $S_t^{j-1}$ we have:

$$\mathbb{E}[\hat{g}_j^t] = \sum_{\substack{i=1 \\ i \notin S_t^{j-1}}}^{N} q_i^{t,j} \left[ \frac{1}{q_i^{t,j}} g_i^t + \sum_{k \in S_t^{j-1}} g_k^t \right]$$

$$= \sum_{\substack{i=1 \\ i \notin S_t^{j-1}}}^{N} g_i^t + \left( \sum_{k \in S_t^{j-1}} g_k^t \right) \underbrace{\left( \sum_{\substack{i=1 \\ i \notin S_t^{j-1}}}^{N} q_i^{t,j} \right)}_{=1}$$

$$= \sum_{i=1}^{N} g_i^t$$

$$= g^t$$

Taking expectation with respect to the choice of $S_t^{j-1}$, and taking the average over $j \in [m]$ we get the result.

(b) We have:

$$\mathbb{E}\left[\|\hat{g}_b^t - g^t\|_2^2\right] = \mathbb{E}\left[\left\|\frac{1}{m}\sum_{j=1}^{m}\hat{g}_j^t - g^t\right\|_2^2\right]$$

$$= \frac{1}{m^2}\sum_{j=1}^{m}\mathbb{E}\left[\|\hat{g}_j^t - g^t\|_2^2\right] + \frac{2}{m^2}\sum_{\substack{j,i \\ j<i}}^{m}\mathbb{E}\left[\langle\hat{g}_j^t - g^t, \hat{g}_i^t - g^t\rangle\right]$$

To show the claim, it is therefore enough to show that second term is zero. Let $j \in [m-1]$. Conditional on $S_t^{i-1}$ we have:

$$\mathbb{E}\left[\langle\hat{g}_j^t - g^t, \hat{g}_i^t - g^t\rangle\right] = \langle\hat{g}_j^t - g^t, \mathbb{E}\left[\hat{g}_i^t\right] - g^t\rangle = 0$$

where we used the fact that the conditional expectation is zero by part (a). Taking expectation with respect to $S_t^{i-1}$ on both sides yields the result.

(c) As discussed in the previous section, (b) applies to all estimators in the class $\mathcal{C}$, so we have for all such estimators:

$$\mathbb{E}\left[\|\hat{g}_b^t(p^{t,1},\ldots,p^{t,m}) - g^t\|_2^2\right] = \frac{1}{m^2}\sum_{j=1}^{m}\mathbb{E}\left[\|\hat{g}_j^t(p^{t,j}) - g^t\|_2^2\right]$$

minimizing over $(p^{t,1},\ldots,p^{t,m})$ by minimizing each term with respect to its variable we get:

$$\operatorname*{argmin}_{(p^{t,1},\ldots,p^{t,m})}\{\mathbb{E}\left[\|\hat{g}_b^t(p^{t,1},\ldots,p^{t,m}) - g^t\|_2^2\right]\} = \left(\frac{p^{t,*}}{1 - \sum_{k=1}^{j-1}p_{I_t^k}^{t,*}}\right)_{j=1}^{m}$$

where:

$$p^{t,*} = \operatorname*{argmin}_{p\in\Delta}\{c_t(p)\} = \frac{\|g_i^t\|_2}{\sum_{j=1}^{N}\|g_j^t\|_2}$$

Recalling that our estimator is in $\mathcal{C}$, and noticing that the optimal probabilities over $\mathcal{C}$ are feasible for our estimator we get the result.

(d) Fix $t \in [T]$. We drop the superscript $t$ from $p^t$, $q_i^{t,j}$, and $g_i^t$. We also drop the subscript $t$ from $I_t^j$ and $S_t^j$ to simplify notation. Define for $j \in [m]$:

$$x_j := \frac{1}{q_{I^j}^j}g_{I^j} \qquad \mu_j := g^t - \sum_{k=1}^{j-1}g_{I^k}$$

We have from part (a):

$$\mathbb{E}\left[x_j\right] = \mu_j$$

Before proceeding with the proof, we first derive an identity relating $q_i^{j+1}$ and $q_i^j$:

$$\frac{1}{q_i^{j+1}} = \frac{1 - \sum_{k\in S^j}p_k}{p_i}$$

$$= \frac{1 - \sum_{k\in S^{j-1}}p_k - p_{I^j}}{p_i}$$

$$= \left(1 - \frac{p_{I^j}}{1 - \sum_{k\in S^{j-1}}p_k}\right)\left(\frac{1 - \sum_{k\in S^{j-1}}p_k}{p_i}\right)$$

$$= \left(1 - q_{I^j}^j\right)\frac{1}{q_i^j}$$

Now, conditional on $S_t^j$, we have:

$$\mathbb{E}\left[\left\|\hat{g}_{j+1}^t - g^t\right\|_2^2\right]$$

$$= \mathbb{E}\left[\left\|x_{j+1} - \mu_{j+1}\right\|_2^2\right]$$

$$= \mathbb{E}\left[\left\|x_{j+1}\right\|_2^2\right] - \left\|\mu_{j+1}\right\|_2^2$$

$$= \left(\sum_{\substack{i=1 \\ i \notin S^j}}^N \frac{1}{q_i^{j+1}} \left\|g_i\right\|_2^2\right) - \left\|\mu_{j+1}\right\|_2^2$$

$$= \left(\sum_{\substack{i=1 \\ i \notin S^{j-1}}}^N \frac{1}{q_i^{j+1}} \left\|g_i\right\|_2^2\right) - \frac{1}{q_{I^j}^{j+1}} \left\|g_{I^j}\right\|_2^2 - \left\|\mu_j - g_{I^j}\right\|_2^2$$

$$= \left(1 - q_{I^j}^j\right) \left(\sum_{\substack{i=1 \\ i \notin S^{j-1}}}^N \frac{1}{q_i^j} \left\|g_i\right\|_2^2\right) - \left(1 - q_{I^j}^j\right) \frac{1}{q_{I^j}^j} \left\|g_{I^j}\right\|_2^2 - \left(\left\|\mu_j\right\|_2^2 - 2\langle\mu_j, g_{I^j}\rangle + \left\|g_{I^j}\right\|_2^2\right)$$

$$= \left(1 - q_{I^j}^j\right) \left(\sum_{\substack{i=1 \\ i \notin S^{j-1}}}^N \frac{1}{q_i^j} \left\|g_i\right\|_2^2 - \left\|\mu_j\right\|_2^2\right) - \left(\frac{1}{q_{I^j}^j} \left\|g_{I^j}\right\|_2^2 - 2\langle g_{I^j}, \mu_j\rangle + q_{I^j}^j \left\|\mu_j\right\|_2^2\right)$$

$$= \left(1 - q_{I^j}^j\right) \mathbb{E}\left[\left\|x_j - \mu_j\right\|_2^2\right] - q_{I^j}^j \left\|x_j - \mu_j\right\|_2^2$$

$$= \left(1 - q_{I^j}^j\right) \mathbb{E}\left[\left\|\hat{g}_j^t - g^t\right\|_2^2\right] - q_{I^j}^j \left\|\hat{g}_j^t - g^t\right\|_2^2$$

where the expectation in the last two lines is conditional on $S^{j-1}$. Taking expectation with respect to $S^{j-1}$ on both sides yields the result.

$\square$

## Appendix D   Extension to constant step-size SGD

While our analysis heavily relies on the assumption of decreasing step-sizes, we have found empirically that a slight modification of our method works just as well when a constant step-size is used. We propose the following epsilon sequence to account for the use of a constant step-size:

$$\varepsilon_t = \frac{1}{C^{1-\delta/3}(C + m(t-1))^{\delta/3}} + p_{min} \tag{23}$$

for a constant $p_{min} \in [0, 1/N]$ and the following condition on $C$:

$$C \le \frac{1}{\frac{1}{N} - p_{min}}$$

which ensures that $\varepsilon_1 \le 1/N$. We ran the same experiment on MNIST, IJCNN1, and CIFAR10 as in Section 6, but with a constant step-size $\alpha_t = \alpha = \frac{m}{2NL}$, and the epsilon sequence (23) with $p_{min} = \frac{1}{5N}$ and $C = \frac{1}{\frac{1}{N} - p_{min}}$, and $\delta = 1$. The results are displayed in figure 3, showing a similar performance compared to the decreasing step-sizes case. Note that choosing a too small $p_{min}$ can start to deteriorate the performance of the algorithm. It is still unclear how to set $p_{min}$ so as to guarantee good performance, but our experiments suggest that setting $\frac{1}{5N}$ is a safe choice.

Figure 3: Comparison of the performance of importance samplers on an l2-regularized softmax regression model on three real world datasets: MNIST (top), IJCNN1 (middle), CIFAR10 (bottom). For this set of experiments, SGD was run using a constant step size.

## Supplementary references

[37] Zalan Borsos, Andreas Krause, and Kfir Y Levy. Online Variance Reduction for Stochastic Optimization. In Sébastien Bubeck, Vianney Perchet, and Philippe Rigollet, editors, *Conference On Learning Theory, COLT 2018, Stockholm, Sweden, 6-9 July 2018*, volume 75 of *Proceedings of Machine Learning Research*, pages 324–357. PMLR, 2018.

[38] Thomas H. Cormen, Charles E. Leiserson, Ronald L. Rivest, and Clifford Stein. *Introduction to Algorithms, 3rd Edition*. MIT Press, 2009.