[Reviews · NeurIPS 2020]

Review 1

Summary and Contributions: The paper investigates the use of adaptive importance sampling in the stochastic gradient descent algorithm for empirical risk type of functions. It focuses on the expected dynamic regret with decreasing step-size. The paper is mainly methodological and the contributions are: - a novel algorithm and its theoretical study regarding the dynamic regret - a variant of their algorithm with mini-batch estimates - some numerical experiments

Strengths: (S1) - The use of an adaptive strategy with decreasing steps for the analysis of dynamic regret seems relatively new. (S2) - Some work has been done to propose an efficient algorithm.

Weaknesses: (W1) - The contributions of the paper are very close from the one of [12]. Indeed, it takes ideas from [12] to restrict the analysis on a simplex that we can control. Moreover, it doesn't clearly put forward the novelty of the bound on dynamic regret compared to the one of [18]. (W2) The mathematics are not well educated and therefore are difficult to check. For instance, the following sentence serves to define a key quantity in the authors development: ‘’For each i ∈ [N ], denote by hti the last observed gradient of fi at time t, with h1i initialized arbitrarily’’ Unfortunately, it is quite unclear as two readers might understand something different. In the same line of comments, the proof of Lemma 3 is very confusing. A list of equality is given without any comment or hint. Given a reasonable time, I think an average reader can not be able to check its correctness. This is partly due to the definition of hit which is unclear. As a consequence, the mathematical statements should be made more rigorous. (W3) - Section 6 on digital experiments needs to be re-written in depth. The authors announce in their contribution l.60-61 that they want to « identify settings in which it leads to significant improvements » but no rigorous analysis is provided. In particular, here are some major points to be modified: (a) first of all the layout of the figures is not clear and it would be clearer for the reader to have names on the axes of each figure. (b) specify more precisely the framework of the experiments and their contexts, in particular recall the objective function, and it would be good for clarity to specify that ISSGD (only mentioned in the legend of a figure) refers to the suggested method. (c) Why talking about the Langevin method (SGLD) since it is not used in the experiments and is only a simple variant? In this perspective, it would be just as legitimate as comparing with all stochastic gradient descent methods (SAG, SAGA, SVRG,…) (d) The developed method being stochastic, the curves were probably averaged: over how many simulations? (e) Why a choice of batch size of order square root N? Can you give some insights for such a choice ?Additional graphs demonstrating the interest of using a batch would be appreciated (since section 5 presents this variant as better). (f) Why did the authors choose a decreasing step-size for only one figure? The reader would like to compare the interest of using an adaptive step (the method proposed by the authors) versus a constant step-size. Moreover, having the constants L and \mu in the choice of the step is not very feasible in practice because these values are generally unknown, in particular the Lipschitz constant, which could possibly be estimated but with a big error leading to a very small step. (e) There is no analysis of the results and no conclusion as to the practicality of using the ISSGD method. It would be valuable to provide insights to explain the behaviour of the algorithm and the results obtained. References: [12] Zalan Borsos, Andreas Krause, and Kfir Y Levy. Online Variance Reduction for Stochastic Optimization. In Sébastien Bubeck, Vianney Perchet, and Philippe Rigollet, editors, Conference On Learning Theory, COLT 2018, Stockholm, Sweden, 6-9 July 2018, volume 75 of Proceedings of Machine Learning Research, pages 324–357. PMLR, 2018. [16] Aryan Mokhtari, Shahin Shahrampour, Ali Jadbabaie, and Alejandro Ribeiro. Online optimization in dynamic environments: Improved regret rates for strongly convex problems. In 55th IEEE Conference on Decision and Control, CDC 2016, Las Vegas, NV, USA, December 12-14, 2016, pages 7195–7201. IEEE, 2016. [18] Lijun Zhang, Shiyin Lu, and Zhi-Hua Zhou. Adaptive Online Learning in Dynamic Environments. In Samy Bengio, Hanna M Wallach, Hugo Larochelle, Kristen Grauman, Nicolò Cesa-Bianchi, and Roman Garnett, editors, Advances in Neural Information Processing Systems 31: Annual Conference on Neural Information Processing Systems 2018, NeurIPS 2018, 3-8 December 2018, Montréal, Canada, pages 1330–1340, 2018.

Correctness: (C1) - Appendix of Lemma 2, line.504 : The constant in the third line is a G squared and not a G as written. This has to be changed, as well as the results that follow from this lemma. The empirical methodology is not convincing enough: (C2) - The numerical details of the experiments are not sufficiently specified (number of simulations, etc.). (C3) - There is a lack of comparisons between constant and adaptive steps to measure the performance of the proposed approach. (C4) - There is no experiment showing the impact of a different batch size or strong convexity.

Clarity: The paper is readable but it suffers from a lack of clarity that increases as it is read: (CL1) - The mention of SGLD, which seems important from the beginning of the paper, is not studied in the experiments and does not bring much to the paper. (CL2) - The very definition of dynamic regret does not make clear the dependence on the past. It would be helpful to clarify this as in Equation (3) of [18]. (CL3) - Subsection 3.3 would probably be more appropriate in the form of an algorithm or pseudo-code. (CL4) - The different assumptions in section 4 are listed one after the other and it would be more pleasant to have some comments and discussions between each one to better appreciate their links. In particular the authors should discuss the links of assumption 3 and the different path-length assumptions in the literature (mentioned in section 2.2 of [18]) (CL5)- Section 6 on numerical experiments needs to be reviewed (see previous remark W3)

Relation to Prior Work: Section 2 discusses in depth past work on online learning problems in the study of static or dynamic regret. The authors are inspired by the approach of [12] and show a similar bound for dynamic regret with a complexity in O(T^{2/3}). It would be a good idea to discuss the different assumptions related to assumption 3 about the behaviour of the trajectory, as this assumption about the path-length is the key to the whole analysis. A huge litterature about sampling gradients is given by the authors in the introduction ([3] to [9]). It is claimed that they are not concerned with any regret analysis but are the methods developed in these references really different from the proposal of the authors. More generally the authors should take care of differentiating their methodological contribution form their theoretical contribution.

Reproducibility: Yes

Additional Feedback: The framework of the paper is asymptotic. In the mean time, the assumptions on the decrease of the step sizes is stated with constants which then contaminates heavily the proofs and statements. This is not realy logic and I suggest to define the step size at 1 / t^delta (starting with t = 1) or A/t^delta. However, if a constant is introduce, then I suggest to add a discussion around its choice/meaning. Below Lemma 3, a confusing discussion takes place about a positive and negative term. It is difficult the understand the presentation choice of the authors regarding Assumption 3 and Proposition 1 (which is trivial). Assumption 3 is valid under the framework of the authors hence useless. If the authors want some gain in the generality of their resutls then it should be more carefully explained. The links with (strong) convexity are not sufficiently discussed. In [16], an important discussion on convexity is addressed and it would be interesting to see where the article fits in this literature. typo line 27 : «approach to achieving » → « approach to achieve » typo line 504 (appendix) : a square appears in a development but should not I encourage the authors to write a new version that would take into consideration my recommendations. [After Rebuttal] I would like to thank the authors for the detailed answer concerning their contributions. As the contribution of the authors appears now more clear to me and seems sufficiently significant (looking at the other reviewers coments), I changed my evaluation accordingly (+1). I still have some concerns about the mathematics of the paper which, from my point of view, are not well educated. For this reason, my final score is 5.


Review 2

Summary and Contributions: This paper provides a new method for importance sampling of examples to use when picking minibatches for SGD. At the t^th iterate, the minibatch is sampled (without replacement) from a sampling distribution p_t over the dataset. A new algorithm is given that leverages the dynamics of SGD to produce sampling distributions p_1,...,p_T for which the sum of the variances estimated gradients is with an additive O(T^{2/3}) of the sum of the variances for the estimates gradients obtainable using the optimal sequence of distributions p_1*,..,p_T^* (i.e. the dynamic regret is O( T^{2/3} ). The technique is pleasingly simple: the algorithm remembers the last observed gradient for each example in the dataset and uses this to construct an estimate of the variance as a function of p_t at the t^th iterate. Many of the last observed gradients are “stale”, but since SGD has decreasing learning rates they cannot be too stale. The algorithm then simple returns the minimizing distribution for this estimate, subject to some minimum weight assigned to each example.

Strengths: The idea and analysis here seem clean and intuitive, and get around several difficulties in past works (e.g. unbounded gradients estimated from bandit information). Getting good dynamic regret is impressive.

Weaknesses: My main concern is that I am not sure how to translate this result into an improved bound for the final optimization algorithm. Is there some realistic scenario in which we can quantify how helpful using this adaptive importance sampling scheme will be vs just uniform samples? Especially given that it looks like each iteration of the importance sampler takes O(N) time and the algorithm requires O(N) memory, I am concerned that I could do better by just doing O(N) iterations of regular SGD with uniform sampling, which takes the same amount of time and less memory. ---- The rebuttal gives me more confidence in the iteration complexity of the algorithm, so I raise my score by one point.

Correctness: The claims seem correct, but I may have missed some details in the appendix.

Clarity: yes

Relation to Prior Work: yes

Reproducibility: Yes

Additional Feedback:


Review 3

Summary and Contributions: The paper proposes an importance sampling strategy for finite sum optimization with decreasing stepsize. The motivation of importance sampling is to reduce the variance of (mini-batch) SGD and the proposed method achieves this in an implementation efficient way. Several aspects of the algorithm are discusses. From a theoretical side the authors show that the algorithm achieves sub-linear dynamic regret (while prior work focused on static regret). And they propose a new unbiased gradient estimator that is compatible with sampling without replacement which is particularly efficient for the proposed method. From a practical side the authors discuss implementation challenges and give pseudo code for the proposed algorithm. Finally, empirical results show the potential of importance sampling on real datasets.

Strengths: I think it is a nice, well written and quite complete piece of work. The proposed algorithm is well motivated and both theoretical as well as practical aspects of the method are addressed. The proof sketch is insightful and the discussion around the implementation challenges helps a practitioner to efficiently implement the proposed method.

Weaknesses: The experimental section is not as well motivated as the rest of the paper, and misses some details.

Correctness: There are some small errors in the theorem statements and the proofs (see comments) but they can easily be fixed and have no major implications on the results.

Clarity: The paper is well written and I particularly like that almost any design choice and assumption is discussed and well motivated.

Relation to Prior Work: I like the related work section a lot. It is not too extensive but points out and explains the issues of existing methods and tools and why things need to be addressed differently. The only thing I miss is an empirical comparison to related methods.

Reproducibility: Yes

Additional Feedback: I enjoyed reading the paper. The first 5 sections are very comprehensive and do not leave many questions unanswered. I like that you investigate the theoretical as well as the practical dimension of your algorithm. I have only a few questions and I found some issues with the analysis that should be fixed. I also have a few suggestions of how to improve the experimental section. 1) You explain how the regret bounds of your method compare to other importance sampling methods, but I am not clear whether this is just a matter of deriving better upper bounds or if your method is indeed superior in practice. Since there is no empirical comparison it is hard for the reader to judge. 2) [247] I understand that a smaller C induces a larger epsilon, but how does this help to overcoming a bad initialization? Is it because you are increasing the probability of sampling unseen points? In any case, I guess sampling is more important since you first need to sample point to change the respective value in memory in any case. 3) Just a as a thought… I wonder if your method could also be used for limited memory training of NNs. Since you get importance measures of datapoints you might benefit from loading them selectively into GPU memory instead of using pre-defined batches (in the spirit of [1]) [1] Efficient Use of Limited-Memory Accelerators for Linear Learning on Heterogeneous Systems, NeurIPS, 2017. 4) Comments and suggestions for experimental section: a) related to question 1: The experiments show that IS leads to significant improvement, but I guess it is not only your method that can achieve this gain. How does it compare to other methods? Since you are using a constant stepsize the is quite a pressing questions because I think also other methods could be applied here. b) the choice of stepsize schedule should be better motivated and discussed. Throughout the paper we focus on decreasing stepsizes and why are you using a fixed stepsize for the experiments? Is it the same schedule for both methods or do you tune it for SGD? c) I thing what could add most value to the paper is an additional figure showing how the convergence changes with the batchsize. This would answer some questions the I had while reading the paper. In particular, is there a lower bound for which the method gets inefficient because the h_i terms are too outdated? For small batch sizes there could be a disadvantage in early rounds because of bad initialization, but how pronounced is this in a practical use-case? d) Maybe also a reference curve using the most recent h_i values for sampling would be interesting. It could illustrate the potential of importance sampling and show how close you get with your method. 5) Comments on theoretical results a) In the statement of Lemma 1 the non-negativity assumption on the numbers should apply to a_i and not a_i^2. The latter does not imply the former and I think it is not correct the way it is now. The same holds for Proposition 2. I would suggest adaption the formulation from the appendix. b) In Proposition 3 you should clarify what the expectation is over 6) Appendix: I did a full pass through the appendix and have a few comments that should be fixed. But overall the proofs are easy to follow and the pseudo code is helpful. a) In the proof of Lemma 2 (line 504) there should be no square in the second term after the second equality. However it has no implications for the rest of the proof. b) In the statement of Lemma 4 the definition of phi is missing and it is not clear what C is. Is it an arbitrary constant? c) In the pseudo code the capitalization of the function names in their definition but not in the call is a bit confusing. In particular for the delete call in line 8 that I assume is not referring to the DELETE procedure. And the statement 32 could be inside the else block for saving a few flops. 7) Minor Comments - Some words should be capitalized throughout the paper. In particular Section, Assumption, Theorem, Lemma, etc, whenever it is followed by a number. Also Euclidean in 33. - The label ISSGD should be introduced in the experimental description - 214 there should be a reference to Proposition 2 ================================================ AFTER REBUTTAL ================================================ I acknowledge that I have read the rebuttal. The authors promised to extend the experimental section, investigate the impact of batch size and learning rate, and add additional baselines to their experiments. This will address most of my concerns and I will stay with my positive score. I additionally encourage the authors to clarify the sections/statements pointed out by the other reviewers and fix the above issues (5,6,7) in the theoretical statements (+ pseudo code).


Review 4

Summary and Contributions: In this work, authors propose importance sampling to reduce the variance in stochastic gradient. The distribution over examples is learned by an online learning method. Estimating the probability is essential for importance sampling, the proposed method seems simple yet effective. For stable learning, they lower bound the probability and decay it when training.

Strengths: Importance sampling is a popular technique to reduce the variance. Authors provide the theoretical analysis about convergence and the experiments for demonstration.

Weaknesses: My concerns are as follows. 1. The regret in [1] is defined on function value while this work defines it with the norm of gradient. It is better to provide the same measurement for a fair comparison. 2. The topic that reduces variance with importance sampling is not new. Besides vanilla SGD, more baselines with variance reduction or importance sampling should be included in experiments for demonstration, especially when the theoretical improvement is not significant. 3. It lacks the experiments for the main method in this work that the gradient is computed from a single example. The only experiment is for mini-batch estimator while the additional experiment for the extreme case is necessary for evaluating the effectiveness of the theorem. 4. Authors claim that the regret bound for the proposed mini-batch method is cast to appendix. However, I didn’t find the regret bound for the mini-batch estimator in the supplementary. [1] Zalan Borsos, Andreas Krause, and Kfir Y Levy. Online Variance Reduction for Stochastic Optimization.

Correctness: seems right

Clarity: Yes

Relation to Prior Work: Yes

Reproducibility: Yes

Additional Feedback: I have read the rebuttal and my major concern can be addressed if they have the promised experiments in the revised version.

[Author Response · NeurIPS 2020]

**Reviewer #4:** Thank you for a detailed review. Regarding your comments on the experiments section, please see the end of this page. We appreciate your comments on the presentation and we will consider incorporating your suggestions in the final version. In terms of content however, we believe that our contributions have been mischaracterized.

**Q**: "The contributions of the paper are very close from the one of [12]"

**A**: We respectfully strongly disagree. The authors in [12] propose an algorithm following the FTRL scheme, our algorithm is based on minimizing newly proposed surrogate cost functions. They study static regret, we study dynamic regret. They use a generic probabilistic bound at the core of their analysis, we use the specific dynamics of SGD/SGLD.

**Q**: "Novelty of the bound on dynamic regret compared to the one of [18]"; "The authors should discuss the links of assumption 3 and the different path-length assumptions in the literature"; "The links with (strong) convexity are not sufficiently discussed"

**A**: The regret bound derived in [18] assumes boundedness of the gradients of the cost functions which does not hold for our problem. Furthermore, our assumptions do no imply any of the path-length assumptions in [18]. Our bound and theirs are therefore not comparable. Besides, the work in [18] deals with the general online learning problem, whereas we are only concerned with the specific problem of adaptive importance sampling for SGD/SGLD. There is not much to be said about (strong) convexity apart from mentioning that our cost functions are convex but not strongly convex, which adds to the difficulty of our problem. The whole point of our work is to show that we are able to achieve a good dynamic regret bound even though most of the commonly used assumptions to study dynamic regret do not hold.

**Q**: "Why talking about the Langevin method (SGLD) ... it would be just as legitimate as comparing with all stochastic gradient descent methods (SAG, SAGA, SVRG,...)"

**A:** It is precisely because the only other method aside from SGD for which our algrorithm/analysis work is SGLD that we mention it. In particular, the variance of the SAG/SAGA/SVRG estimators has a completely different form, and designing adaptive importance sampling techniques for these estimators remains an open problem. Furthermore there has been a lot of work in applying variance reduction techniques initially designed for optimization to sampling (see e.g. [35]), and we view our work as a contribution to that literature as well.

**Reviewer #6:** Thank you for an in-depth read of our paper. We address your concern below.

**Iteration complexity:** At the theoretical level, we can show better convergence rates by replacing our dynamic regret bound directly in the standard convergence bound of averaged SGD as is done in ([11], Theorem 3 and Corollary 1). This does not cover the last-iterate case however. Since we are able to obtain per-step regret guarantees (Lemma 2 + Lemma 4), we can also obtain bounds on a weighted version of the dynamic regret, which allows us to obtain improved last-iterate guarantees for SGD compared to uniform sampling. We will consider including these results in a new section in the appendix if the reviewer thinks it would be a good addition. In practice, the magnitude of the improvement depends on how much variance reduction can be achieved through the use of the optimal probabilities vs uniform sampling. This is what we addressed in the experiments section, where we mentioned over-parametrized models and imbalanced datasets as examples of cases in which this improvement is significant. **Cost per iteration:** We would like to point out that while our algorithm is $\mathcal{O}(N)$ in both time and space, it has no dimension dependence. For large scale high-dimensional problems, the cost of the final optimization algorithm will still be largely dominated by the gradient evaluations, and the overhead from our sampler will be negligible. The fact that the memory cost is independent of the dimension of the problem is what makes our algorithm an attractive alternative to control-variate methods like SAG/SAGA/SVRG.

**Reviewer #7:** Thank you for your very positive comments and helpful suggestions! We will incorporate your suggestions for the experiments section in the final version, please see the paragraph below.

**Reviewer #8**: Thank you for taking the time to read our paper.

**Concern 1**: Their framework is slightly more general than ours, but when specialized to importance sampling for SGD it becomes exactly the same. Please see the last two paragraphs of the introduction in their paper.

**Concerns 2, 3**: See the discussion below. We disagree however on the significance of the theoretical improvement.

**Concern 4**: This was a mistake. Please see the supplementary version of the paper.

**Comments on the experiments section:** We would like to emphasize that our aim was not to provide a comprehensive empirical study of our algorithm, but only to illustrate its performance on a few datasets, and point to cases in which the improvement is substantial. This is inevitable given the space constraint. That being said, we agree with the reviewers that this section of our work can be improved, and we plan on making full use of the additional page in the final version to do so. In particular we plan on: (i) Giving more details on the setup of the experiments. (ii) Include an experiment that explores more the effect of the choice of step-size (decreasing vs constant) and batch size on the performance of the algorithm. This will include experiments with decreasing step-sizes and a batch-size of 1 which corresponds to the setting of our regret bound. (iii) Include results from other baselines in all of the experiments. (iv) Adding a paragraph analyzing the results and more explicitly explaining when and why the algorithm leads to significant improvements.

[Meta-Review · NeurIPS 2020]

The paper has received positive reviews and the author response addressed the concerns of the reviewers sufficiently. Thus, the paper is deemed suitable for publication for NeurIPS, although I strongly urge the authors to make the changes that they have promised in their rebuttal.